# LEARNING SHARED MANIFOLD REPRESENTATION OF IMAGES AND ATTRIBUTES FOR GENERALIZED ZERO-SHOT LEARNING

## ABSTRACT

Many of the zero-shot learning methods have realized predicting labels of unseen images by learning the relations between images and pre-defined class-attributes. However, recent studies show that, under the more realistic generalized zero-shot learning (GZSL) scenarios, these approaches severely suffer from the issue of *biased* prediction, i.e., their classifier tends to predict all the examples from both seen and unseen classes as one of the seen classes. The cause of this problem is that they cannot properly learn a mapping to the representation space generalized to the unseen classes since the training set does not include any unseen class information. To solve this, we propose a concept to learn a mapping that embeds both images and attributes to the shared representation space that can be generalized even for unseen classes by interpolating from the information of seen classes, which we refer to *shared manifold learning*. Furthermore, we propose *modality invariant variational autoencoders*, which can perform shared manifold learning by training variational autoencoders with both images and attributes as inputs. The empirical validation of well-known datasets in GZSL shows that our method achieves the significantly superior performances to the existing relation-based studies.

## 1 INTRODUCTION

The recent high performance of deep neural networks on image classification and object recognition depends greatly on whether we can obtain sufficiently labeled images of classes to predict. However, it is difficult to operate such a AI system as-is in the real world because the number of existing classes is huge and, as long as human beings create or develop new objects, their number can be expected to continue to increase day by day, resulting in difficulty obtaining labeled data of all these classes. Human beings have the excellent ability to infer objects that they have seen only a few by exploiting and transferring the knowledge learned in the past so as to survive in the harsh real world. In the machine learning context, this problem is formulated as few-shot learning.

The extreme problem setting of few-shot learning is especially known as zero-shot learning (ZSL), which is to train by labeled set from certain classes called *seen* classes, and then to predict completely *unseen* classes not included in the training set. It is usually accomplished by preparing pre-defined semantic representations of all classes such as attributes, and learning the relations between images and class-attributes, which we call the relation-based method in this paper (Lampert et al., 2009; 2014; Frome et al., 2013; Akata et al., 2013; 2015; Romera-Paredes & Torr, 2015; Kodirov et al., 2017). However, this conventional setting is unnatural in application to the real world because the objects we encounter are not always new but are rather likely to have been experienced in the past. Therefore, a more natural setting that the test set includes examples of both seen and unseen classes has been proposed, which is called generalized zero-shot learning (GZSL) (Chao et al., 2016).

It is known that GZSL is a more difficult setting than conventional ZSL because seen classes are also candidates for prediction. Especially, it is pointed out that prediction to unseen classes becomes fail severely in the case of relation-based methods (Chao et al., 2016; Xian et al., 2018). The main reason is that they cannot obtain the generalized representation space such that examples of unseen classes are properly mapped because this information is not included in the training set. For example,

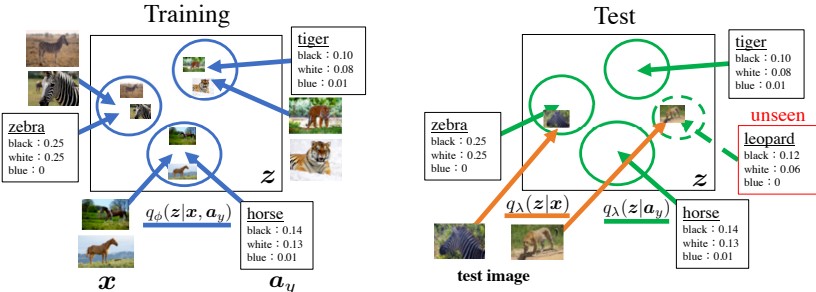

Figure 1: An overview of shared manifold learning with MIVAE to achieve GZSL. At training time, MIVAE learns a mapping to latent space $z$, $q_\phi(z|x, a_y)$, so that it integrates image $x$ and class-attributes $a_y$, and properly places even unseen data. Note that mappings of each of the image and the attributes, $q_\lambda(z|x)$ and $q_\lambda(z|a_y)$, are also learned to approximate this mapping, $q_\phi(z|x, a_y)$. At test time, we can properly arrange the attributes of the unseen classes on the latent space using the learned mapping of the attributes, $q_\lambda(z|a_y)$. Therefore, using the learned mapping of the image, $q_\lambda(z|x)$, the test image can be embedded in the vicinity of the correct class-attributes.

if unseen classes are arranged to overlap with the seen classes in the representation space, which means this space is not generalized for unseen classes, all examples of the unseen classes might be *biased* to be predicted as one of the seen classes.

Therefore, in this study, we propose a new solution of GZSL called *shared manifold learning*. At first, we aim to learn a mapping that embeds each image and attribute in the same representation space, i.e., *shared representation learning* of images and attributes. This contributes to the prediction of the relations in shared representation that is robust against the representation of the inputs. In addition, we make sure that this mapping is also generalized, which means that the inputs of the unseen classes are embedded in appropriate places in shared representation by interpolating from the information of seen classes. That is, it can be regarded as *manifold learning* in shared representation. If such shared manifold learning can be performed properly, we can estimate the correct class-attributes from the test image by embedding it in the shared manifold representation, regardless of whether this image is seen or unseen.

To realize this, we propose modality invariant variational autoencoders (MIVAE), which are based on variational autoencoders (VAE) (Kingma & Welling, 2013; Rezende et al., 2014). In this method, first of all, to obtain shared representations from inputs of images $x$ and class-attributes $a_y$, we define mappings from each input to the latent space $z$ as the deep probabilistic distributions, $q_\lambda(z|x)$ and $q_\lambda(z|a_y)$. Next, to learn manifold representation from these two inputs, we prepare a VAE with both images and attributes as inputs. In learning of MIVAE, we not only maximize the lower bound of this VAE but also learn so that each mapping approximates the encoder of VAE, $q_\phi(z|x, a)$, and so that each mapping also approaches another. This makes it possible to obtain the mappings that embed each input in the shared manifold representation. See figure 1 as the outline of shared manifold learning with MIVAE. If we consider images and attributes as different *modalities*, then MIVAE can be regarded as an extension of works examining multimodal VAE (Suzuki et al., 2016; Vedantam et al., 2017).

The contribution of this research is as follows.

- We discuss that the biased problem is related to the reason why the relation-based methods in GZSL fails and propose shared manifold learning as a concept to solve it.

- We propose MIVAE, which extended VAE as a method to realize shared manifold learning.

- We confirm that MIVAE can perform shared manifold learning properly and that it can contribute to the higher performance of GZSL compared to existing relation-based GZSL methods.

## 2 PROBLEM FORMULATION

We assume that the dataset $\mathcal{D}_{tr} = \{x_i, y_i\}_{i=1}^{N_{tr}}$ is given as the training set, where $x_i \in \mathcal{X}$ is the input data, e.g. an image, and $y_i \in \mathcal{Y}_s = \{1..., S\}$ is the corresponding label data. The objective

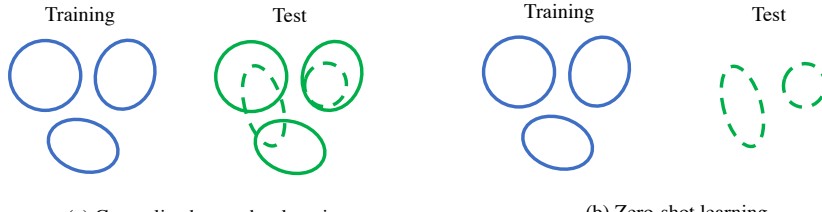

(a) Generalized zero-shot learning  (b) Zero-shot learning

Figure 2: Explanation of biased problem in GZSL. Each circle represents the distribution in the representation space of each class, and the dotted circles are the unseen classes. In GZSL, if the representations of the unseen classes overlap the seen classes, all unseen data might be predicted in any of the seen classes. On the other hand, ZSL does not have such a problem as there are no seen classes at test time. Note that in this explanation, we ignore the difficulty of learning the generalized relations between images and attributes, but actually, it is necessary to consider the influence of that as well.

of ZSL is to train the classifier using $\mathcal{D}_{tr}$ and to predict labels $\hat{y}_j \in \mathcal{Y}$ from the example $\boldsymbol{x}_j \in \mathcal{X}$ in the test set $\mathcal{D}_{ts} = \{\boldsymbol{x}_j\}_{j=1}^{N_{ts}}$. In the original ZSL, it is assumed that the classes of the test set are completely unseen in the training set, which means $\mathcal{Y} = \mathcal{Y}_u = \{S+1, ..., U\}$. On the other hand, our goal is to train in the setting of GZSL, which includes both seen and unseen classes in the test set ($\mathcal{Y} = \mathcal{Y}_s \cap \mathcal{Y}_u$).

We also assume that we have the class-attribute matrix $\boldsymbol{A} \in \mathbb{R}^{M \times (S+U)}$ as the semantic information of classes, where each column represents the $M$-dimensional attribute vector $\boldsymbol{a}_c \in \mathcal{A} = \mathbb{R}^M$ of each class $c = 1, ..., S+U$. Using this attribute vector, the training set $\mathcal{D}_{tr}$ can be replaced as $\{\boldsymbol{x}_i, \boldsymbol{a}_{y_i}\}_{i=1}^{N_{tr}}$.

## 3 PROPOSED METHOD

### 3.1 BIASED PROBLEM IN RELATION-BASED GZSL AND SHARED MANIFOLD LEARNING

In this section, we discuss the reason why the prediction of test data fails in relation-based GZSL and how to solve it.

The relation-based methods learn the relations between the image and the attribute by mapping them to the representation space. Depending on each method, this representation space is considered as attribute space (image-attribute projection) or another shared space (shared representation learning). For details, see the section 4.

What is important in learning both ZSL and GZSL is to obtain the relations between images and attributes generalized to test data, that is, to map each pair of examples to the same place in the representation space. Furthermore, in the setting of GZSL, since the test set includes the seen classes, it is necessary to learn a generalized representation where the unseen classes do not overlap with the seen classes. However, this is clearly difficult because information of the unseen classes does not exist in the training set. If the generalization between classes fails and the representations of the unseen classes overlap the seen classes, all unseen examples may be classified biased toward the seen classes (see figure 2(a)). Therefore, in GZSL, we have to consider means to learn generalized representations not only between images and attributes but also between classes. Note that ZSL does not include a biased problem like GZSL because the test set does not include the seen classes (see figure 2(b)).

In this paper, we propose the framework of learning to solve this problem as follows.

First, we consider mapping from each image and attribute to the same space. At this time, we enforce the images and attributes to be the same representations in the embedding space (shared representation learning). One of the advantages of mapping to the shared space is that we can think of relations between examples in a space with more information by integrating images and attributes. It is thought that we can estimate the location of unseen data better by performing manifold learning as described later in such a space. Also, Chao et al. (2016) argues that improvement of the representation of attributes is vital for GZSL, which can be realized by the shared representation integrated with the images. Another advantage is that since shared space does not depend on input dimensions or representation, we can make robust inference to the complexity of inputs.

Next, we perform manifold learning to obtain a shared representation generalized in the unseen classes. In manifold learning, continuous mapping from input space to shared space can be learned. Therefore, unseen data also can be mapped to an appropriate place in the shared space by interpolating from the prior knowledge of the seen classes.

We designate these series of learning as *shared manifold learning* of images and attributes. We claim that we can deal with the bias-related shortcoming of GZSL by learning shared manifold representation.

## 3.2 MULTIMODAL INVARIANT VARIATIONAL AUTOENCODER

Next, we propose modality invariant VAE (MIVAE), which is a method of learning mapping from two inputs to a space to be shared and manifold representations.

Here, we consider that the dataset of ZSL $\{\boldsymbol{x}_i, \boldsymbol{a}_{y_i}\}_{i=1}^{N_{tr}}$ has different *modalities*, which means that $\boldsymbol{x}$ and $\boldsymbol{a}_y$ are different in their spaces and distributions, i.e., $\mathcal{X} \neq \mathcal{A}$ and $p(X) \neq p(A)$, where $X = \{x_1, ..., x_{N_{tr}}\}$ and $A = \{a_{y_1}, ..., a_{y_{N_{tr}}}\}$.

At first, we consider two Gaussian distributions parameterized with deep neural networks $q_\lambda(\boldsymbol{z}|\boldsymbol{x})$ and $q_\lambda(\boldsymbol{z}|\boldsymbol{a}_y)$ as mappings from each of $\mathcal{X}$ and $\mathcal{A}$ to $\mathcal{Z}$, where $\lambda$ is the parameter of the distribution. See appendix A for the way of parameterization with deep neural networks. In MIVAE, we aim to learn so that these mappings embed inputs in shared manifold representation. Note that it is important to set these mappings to the deep probabilistic distributions in obtaining shared representations. For a detailed discussion on this, see appendix C.

Next, we use variational autoencoders (VAEs) (Kingma & Welling, 2013; Rezende et al., 2014), which has been widely used as a model of representation learning and manifold learning. Moreover, we consider VAE with two modalities as input, which is called joint VAE (Vedantam et al., 2017).

Joint VAE has a latent concept $\boldsymbol{z}$ that is common to the two modalities $\boldsymbol{x}, \boldsymbol{a}_y$, each of which is generated conditionally and independently from the concept. That is, their generative processes are $\boldsymbol{z} \sim p(\boldsymbol{z})$ and $\boldsymbol{x}, \boldsymbol{a}_y \sim p_\theta(\boldsymbol{x}, \boldsymbol{a}_y|\boldsymbol{z}) = p_\theta(\boldsymbol{x}|\boldsymbol{z})p_\theta(\boldsymbol{a}_y|\boldsymbol{z})$, where $\theta$ corresponds to model parameters. These processes indicate that the latent variable $\boldsymbol{z}$ can be regarded as a shared representation containing different modality information.

To train this, we should maximize the joint distribution $p(\boldsymbol{x}, \boldsymbol{a}_y) = \int p_\theta(\boldsymbol{x}, \boldsymbol{a}_y|\boldsymbol{z})p(\boldsymbol{z})d\boldsymbol{z}$ over the training set. However, it is difficult to perform this maximization directly because this joint distribution is intractable. Therefore, we instead maximize the following evidence lower bound (ELBO).

$$\mathcal{L}_{vae} = -D_{KL}(q_\phi(\boldsymbol{z}|\boldsymbol{x}, \boldsymbol{a}_y)||p(\boldsymbol{z})) + E_{q_\phi(\boldsymbol{z}|\boldsymbol{x}, \boldsymbol{a}_y)}[\log p_\theta(\boldsymbol{x}|\boldsymbol{z})] + E_{q_\phi(\boldsymbol{z}|\boldsymbol{x}, \boldsymbol{a}_y)}[\log p_\theta(\boldsymbol{a}_y|\boldsymbol{z})], \quad (1)$$

where $q_\phi(\boldsymbol{z}|\boldsymbol{x}, \boldsymbol{a}_y)$ is an approximate distribution of posterior and $\phi$ represents parameters. We call $q_\phi(\boldsymbol{z}|\boldsymbol{x}, \boldsymbol{a}_y)$ as encoder and $p_\theta(\boldsymbol{x}|\boldsymbol{z})$ and $p_\theta(\boldsymbol{a}_y|\boldsymbol{z})$ as decoder. These models are also parameterized by deep neural networks. The distribution of the encoder is Gaussian, and that of the decoder depends on the distribution of the input data.

From the encoder of trained VAE, we can extract shared representations that integrate the information of different modalities. In addition, the VAE can obtain the manifold representation of the input in the latent space and appropriately place the unseen input on the latent space by interpolating the seen information. Because we use images and attributes as inputs to joint VAE here, we can obtain the encoder that integrates and maps them into manifold representation, i.e., *we can perform shared manifold learning*.

Therefore, if we train each modality's distribution to minimize the distance between the encoder, then we can obtain such representations from the distribution of each modality. We define this distance using Kullback-Liebler (KL) divergence as

$$\mathcal{L}_{dist} = D_{KL}(q_\phi(\boldsymbol{z}|\boldsymbol{x}, \boldsymbol{a}_y)||q_\lambda(\boldsymbol{z}|\boldsymbol{x})) + D_{KL}(q_\phi(\boldsymbol{z}|\boldsymbol{x}, \boldsymbol{a}_y)||q_\lambda(\boldsymbol{z}|\boldsymbol{a}_y)). \quad (2)$$

Intuitively, because the encoder has more information to be conditioned, it seems that the mapping distributions, especially $q_\lambda(\boldsymbol{z}|\boldsymbol{a}_y)$, might degenerate if each is brought closer to the encoder. Actually, Vedantam et al. (2017) showed that minimization of $E_{p_{data}(\boldsymbol{x}, \boldsymbol{a}_y)}[D_{KL}(q_\phi(\boldsymbol{z}|\boldsymbol{x}, \boldsymbol{a}_y)||q_\lambda(\boldsymbol{z}|\boldsymbol{a}_y))]$ is equivalent to minimization of

Table 1: Difference in approaches of ZSL.

| Method | Type | Training cost | Complex data | Evading biased problem in GZSL |
|---|---|---|---|---|
| Shared manifold learning | Relation-based | Low | Yes | Yes |
| Image-attribute projection | Relation-based | Low | Yes | No |
| Shared representation learning | Relation-based | Low | Yes | No |
| Image synthesis | Synthesis-based | High | No | Yes |

$E_{p_{data}(\boldsymbol{x},\boldsymbol{a})}[D_{KL}(q_\phi^{avg}(\boldsymbol{z}|\boldsymbol{a}_y)||q_\lambda(\boldsymbol{z}|\boldsymbol{a}_y))]$, where $q_\phi^{avg}(\boldsymbol{z}|\boldsymbol{a}_y) = E_{p_{data}(\boldsymbol{x}|\boldsymbol{a}_y)}[q_\phi(\boldsymbol{z}|\boldsymbol{x},\boldsymbol{a}_y)]$, i.e., this is the average of the encoder over all $\boldsymbol{x}$ corresponding to $\boldsymbol{a}$. Therefore, the minimization of equation 2 encourages each mapping to close to the distribution complementing the other modality of the encoder, thereby resulting in stabilization.

When equation 2 becomes minimum, both $q_\lambda(\boldsymbol{z}|\boldsymbol{x}) = \int q_\phi(\boldsymbol{z}|\boldsymbol{x},\boldsymbol{a})d\boldsymbol{a}$ and $q_\lambda(\boldsymbol{z}|\boldsymbol{a}) = \int q_\phi(\boldsymbol{z}|\boldsymbol{x},\boldsymbol{a})d\boldsymbol{x}$ are satisfied. However, if the objective function of VAE is also optimized simultaneously, then these minimization may not be performed properly due to the influence of the VAE terms during training. Therefore, to force the two distributions to be explicitly close at the training time, we also minimize the following equation:

$$\mathcal{L}_{map} = D_{KL}(q_\lambda(\boldsymbol{z}|\boldsymbol{x})||q_\lambda(\boldsymbol{z}|\boldsymbol{a}_y)). \tag{3}$$

Therefore, the objective function of MIVAE is

$$\mathcal{L} = -\mathcal{L}_{vae} + \lambda_{dist}\mathcal{L}_{dist} + \lambda_{map}\mathcal{L}_{map}, \tag{4}$$

where $\lambda_{dist}$ and $\lambda_{map}$ are hyper-parameters. When $\lambda_{map} = 0$, this objective function is the same as JMVAE (Suzuki et al., 2016), which means that this model can be regarded as an extension of JMVAE.

After training MIVAE, a label of test image can be simply predicted as $\hat{y}_j = \arg\min_{y\in\mathcal{Y}} D_{KL}(q_\lambda(\boldsymbol{z}|\boldsymbol{x}_j)||q_\lambda(\boldsymbol{z}|\boldsymbol{a}_y))$.

# 4 RELATED WORKS

ZSL is one of the greatest attention problems among image recognition tasks, and various methods have been proposed. Here, we explain them separately on relation-based and synthetic-based approaches.

## 4.1 RELATION-BASED

Relation-based methods are intended to train relations between images and class-attributes $F(x, a_y)$ from training data and to predict a class whose relation is strong in test data, i.e. $\hat{y} = F(x, a_y)$. This approach can be classified further into the following two methods: training of mapping from image space to attribute space (image-attribute projection) and training of shared representation that embed both image and attribute space (shared representation learning).

**Image-attribute projection** In ZSL, the simplest and most widely adopted approach is to obtain mapping from image space to class-attribute space with training data. Some early studies used classifiers such as SVM to learn mapping from images to class attributes (Lampert et al., 2009; 2014), but many other methods use linear embedding into attributes (or semantic representations) and typically set a ranking loss as the objective function to train such embedding (Frome et al., 2013; Akata et al., 2013; 2015; Romera-Paredes & Torr, 2015; Kodirov et al., 2017). Actually, Romera-Paredes & Torr (2015) adds a regularization term of the mapping to this objective function. Also, Kodirov et al. (2017) minimizes the embedding error in both directions with the objective. Recently, nonlinear models are also used Socher et al. (2013) to consider non-linearity between images and attributes. However, classifiers tend to predict only one class from many images at testing, which is known as the hubness problem because of the difference in the distribution of the two modalities. To avoid this difficulty, some works have proposed training of reverse mapping from attribute to image space with such as deep neural networks (Zhang et al., 2017). Verma & Rai (2017) suggests that mapping from attributes to an image are a conditional distribution by exponential family rather than deterministic, which can incorporate consideration of uncertainty among modalities. Therefore the hubness problem can be relaxed further.

**Shared representation learning** Another means of obtaining relations is to learn mapping that embeds images and attributes into a shared latent representation. This learning can be realized by shared

Table 2: Summary of datasets in our experiments.

| Dataset | Images | Attributes | Seen/Unseen classes |
|---------|--------|------------|---------------------|
| SUN | 14340 | 102 | 150/50 |
| CUB | 11788 | 312 | 645/72 |
| AwA1 | 30475 | 85 | 40/10 |
| AwA2 | 37322 | 85 | 40/10 |
| aPY | 15339 | 64 | 20/12 |

representation learning in which images and attributes are regarded as different modalities. In ZSL, several methods of learning shared embedding spaces have been proposed (Zhang & Saligrama, 2015; 2016), but few methods use deep neural networks as embedding maps. VZSL (Wang et al., 2017) learns shared representations of images and attributes using deep generative models, which most closely approximates our method. As with Verma & Rai (2017), uncertainty can be taken into consideration, and furthermore, deep neural networks are used.

Although these relation-based methods work well in conventional ZSL settings, in GZSL, their performance is much worse, as shown in Chao et al. (2016). The approach of shared representation learning is robust to the representation of attributes, but it does not perform manifold learning at two inputs, which means that it does not perform shared manifold learning. Indeed, VZSL performs manifold learning only with image input, which results in poor generalization performance in examples of unseen classes, as shown later in an experiment.

## 4.2 Synthesis-based

Recently, several synthesis-based methods have been proposed (Verma et al., 2018; Mishra et al., 2017). These approaches are attempts to generate images from class-attributes and to train a classifier of unseen classes by these synthesis samples. To generate samples from attributes properly, they train conditional deep generative models of images given class-attributes in training data. It is apparently difficult to generate realistic test images that are useful to train the classifier. However, in many ZSL settings, it is common to use feature extraction representations from images as input so that they can generate these inputs with high quality.

The greatest advantage of the synthesis-based approaches is that they can achieve high performance in a GZSL setting because they train classification with a new classifier after generating all samples of seen or unseen classes from the trained deep generative model. For that reason, we no longer need to consider differences between seen and unseen classes. However, training the classifier requires generation of multiple images of all classes. Therefore, if the image size increases or the number of classes increases, then it might be costly to generate these synthetic samples. Moreover, this method works well only if the input representations of images are easy to generate. In recent years, several deep generative models have been proposed that can produce high-quality images (Kingma & Dhariwal, 2018; Karras et al., 2017), but it still might be difficult to generate arbitrary types of inputs to be useful for classification of unseen classes.

In this paper, we focused on the relation-based approach and proposed the concept of shared manifold learning to deal with the biased problem while retaining the advantage of relation-based GZSL. Table 1 presents a brief comparison of our method with others.

## 5 Experiments

For experiment, we use the following four datasets commonly used for ZSL: Animals with Attributes (AWA1) (Lampert et al., 2014), CUB-200-2011 Bird (CUB) (Wah et al., 2011), SUN Attribute (SUN) (Patterson & Hays, 2012), and Attribute Pascal and Yahoo (aPY) (Farhadi et al., 2009). In addition, because the images of AWA1 were not public available, Xian et al. (2018) additionally introduced Animals with Attributes 2 (AWA2) composed of images collected from public sites. Therefore, in this experiment, we will use five datasets including AWA2. Each dataset contains images of each class, where each class is represented by semantic attributes. Table 2 presents the numbers of images, attributes, and classes in respective datasets. For fair comparisons, we use 2048-dim top-layer embedding of the 101-layered ResNet (He et al., 2016) provided by Xian et al. (2018) as the image vector. Furthermore, for the class-attribute representation we use the continuous valued attributes between 0 and 1 provided with each dataset. We followed the split proposed in Xian et al. (2018) for splitting each dataset into train, validation, and test. The hyper-parameter selection of

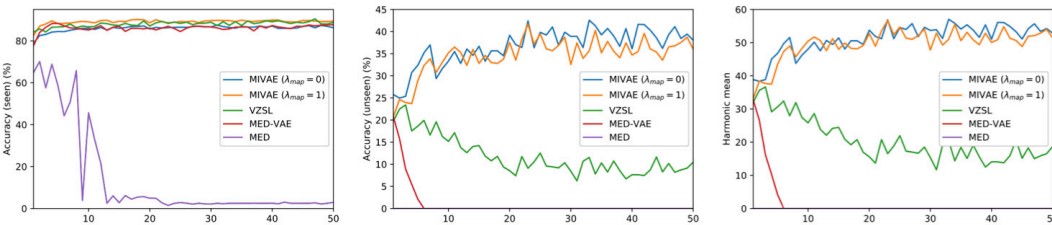

Figure 3: Learning curves in GZSL with joint representation learning methods. The left and middle plot respectively present the accuracy of seen and unseen classes in the test data. The right shows the harmonic mean evaluation.

our model was based on this train-validation split, whereas in training of GZSL both were used as training data.

For the metric of evaluation at the test time, we use the average per-class top-1 accuracy on both the seen and unseen classes (referred as $acc_{\mathcal{Y}_s}$ and $acc_{\mathcal{Y}_u}$). In addition, to evaluate the performance on GZSL, we calculate the harmonic mean of $acc_{\mathcal{Y}_s}$ and $acc_{\mathcal{Y}_u}$, which is $H = (2 \cdot acc_{\mathcal{Y}_s} \cdot acc_{\mathcal{Y}_u})/(acc_{\mathcal{Y}_s} + acc_{\mathcal{Y}_u})$.

We used the Adam optimization algorithm (Kingma & Ba, 2014) with a learning rate of $10^{-3}$. Also, in order to suppress over-fitting, we applied batch normalization to each layer and added L2 regularization to the parameters, where the coefficient of this term is $10^{-4}$. In all experiments, we trained for 200 epochs and evaluated the average of 5 experiments with different seeds. For details of network architectures, see appendix A. In all experiments, we set $\lambda_{dist} = 1$, which works well in multimodal learning according to reports of related studies (Suzuki et al., 2016; Vedantam et al., 2017). For another parameter $\lambda_{map}$, we will verify the effect by changing this value in the experiments. All models in this paper were implemented with PyTorch (Paszke et al., 2017).

We conduct two experiments. First, compared with some methods of shared representation learning, we demonstrate that MIVAE with shared manifold learning can learn with high performance qualitatively and quantitatively in GZSL. The second experiment compares our method with the state-of-the-art relation-based methods and confirms that our method has much higher performance than those methods.

### 5.1 COMPARISON WITH SHARED REPRESENTATION LEARNING METHODS

To the best of our knowledge, VZSL (Wang et al., 2017) is the only existing method of ZSL used to train shared representation learning with deep probabilistic embedding maps, $q_\lambda(\boldsymbol{z}|\boldsymbol{x})$ and $q_\lambda(\boldsymbol{z}|\boldsymbol{a}_y)$. Furthermore, we consider the following three methods including VZSL as shared representation learning to compare.

**Minimization of embedded divergence (MED)** This simply minimizes KL divergence of the two distributions, i.e. the loss function of MED is $D_{KL}(q_\lambda(\boldsymbol{z}|\boldsymbol{x})||q_\lambda(\boldsymbol{z}|\boldsymbol{a}_y))$.

**MED-VAE** To train the manifold representation in the latent space, we add the reconstruction error of the VAE whose input is the image vector to that of the MED, i.e., $-E_{q_\lambda(\boldsymbol{z}|\boldsymbol{x})}[\log p_\theta(\boldsymbol{x}|\boldsymbol{z})] + D_{KL}(q_\lambda(\boldsymbol{z}|\boldsymbol{x})||q_\lambda(\boldsymbol{z}|\boldsymbol{a}_y))$.

**VZSL (Wang et al., 2017)** Wang et al. (2017) proposed margin regularization that encourages separation of the true class from the next class. The loss function of VZSL is the addition of the margin regularization to that of MED-VAE, i.e.,

$$-E_{q_\lambda(\boldsymbol{z}|\boldsymbol{x})}[\log p_\theta(\boldsymbol{x}|\boldsymbol{z})] + D_{KL}(q_\lambda(\boldsymbol{z}|\boldsymbol{x})||q_\lambda(\boldsymbol{z}|\boldsymbol{a}_y)) - \lambda_{vzsl} \log \sum_{c=1}^{S} \exp(-D_{KL}(q_\lambda(\boldsymbol{z}|\boldsymbol{x})||q_\lambda(\boldsymbol{z}|\boldsymbol{a}_c))),$$

where $\lambda_{vzsl}$ is a hyper-parameter and set to 1.

This experiment used AWA2 for the dataset. The distribution of the embedding maps was Gaussian. The networks structures were set the same in all methods. Furthermore, the prediction methods of the classes of each example in the test data are all the same: $\hat{y}_j = \arg\min_{y \in \mathcal{Y}} D_{KL}(q_\lambda(\boldsymbol{z}|\boldsymbol{x}_j)||q_\lambda(\boldsymbol{z}|\boldsymbol{a}_y))$.

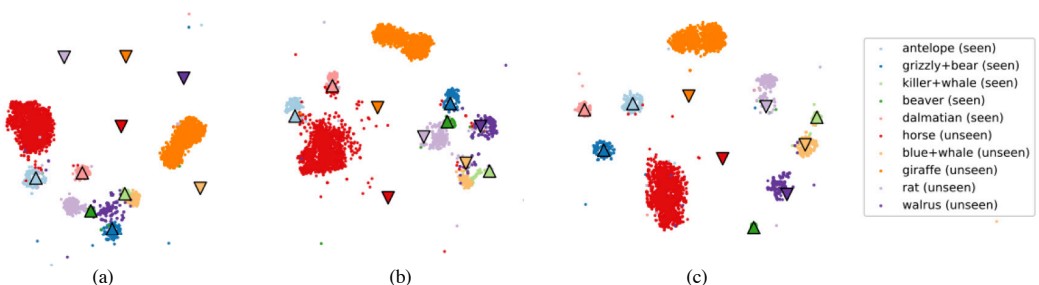

Figure 4: t-SNE plots of joint latent representations. After selecting five seen classes and five unseen classes from AWA2, we embedded their images and attributes. Circle plots are the embedding of the images. Triangles represent embedding of the attributes ($\triangle$ denotes a seen class and $\triangledown$ is an unseen class). Each color of the plot corresponds to a different class: (a) MED-VAE, (b) VZSL, and (c) MIVAE, where $\lambda_{map} = 1$.

Table 3: Results in the GZSL setting. Each of **u** and **s** represents class-mean top-1 accuracy (%) on unseen and seen classes and **H** means the harmonic mean of **u** and **s**. We also show the average of **H** in all datasets.

| Methods | SUN | | | CUB | | | AWA1 | | | AWA2 | | | aPY | | | Average |
| | u | s | H | u | s | H | u | s | H | u | s | H | u | s | H | H |
|---|---|---|---|---|---|---|---|---|---|---|---|---|---|---|---|---|
| CONSE (Szegedy et al., 2015) | 6.8 | 39.9 | 11.6 | 1.6 | 72.2 | 3.1 | 0.4 | **88.6** | 0.8 | 0.5 | **90.6** | 1.0 | 0.0 | **91.2** | 0.0 | 3.3 |
| CMT (Socher et al., 2013) | 8.1 | 21.8 | 11.8 | 7.2 | 49.8 | 12.6 | 0.9 | 87.6 | 1.8 | 0.5 | 90.0 | 1.0 | 1.4 | 85.2 | 2.8 | 6.0 |
| SSE (Zhang & Saligrama, 2015) | 2.1 | 36.4 | 4.0 | 8.5 | 46.9 | 14.4 | 7.0 | 80.5 | 12.9 | 8.1 | 82.5 | 14.8 | 0.2 | 78.9 | 0.4 | 9.1 |
| LATEM (Xian et al., 2016) | 14.7 | 28.8 | 19.5 | 15.2 | 57.3 | 24.0 | 7.3 | 71.7 | 13.3 | 11.5 | 77.3 | 20.0 | 0.1 | 73.0 | 0.2 | 15.4 |
| ALE (Akata et al., 2016) | 21.8 | 33.1 | 26.3 | 23.7 | 62.8 | 34.4 | 16.8 | 76.1 | 27.5 | 14.0 | 81.8 | 23.9 | 4.6 | 73.7 | 8.7 | 24.2 |
| DEVISE (Frome et al., 2013) | 16.9 | 27.4 | 20.9 | 23.8 | 53.0 | 32.8 | 13.4 | 68.7 | 22.4 | 17.1 | 74.7 | 27.8 | 4.9 | 76.9 | 9.2 | 20.3 |
| SJE akata2015evaluation | 14.7 | 30.5 | 19.8 | 23.5 | 59.2 | 33.6 | 11.3 | 74.6 | 19.6 | 8.0 | 73.9 | 14.4 | 3.7 | 55.7 | 6.9 | 18.9 |
| ESZSL (Romera-Paredes & Torr, 2015) | 11.0 | 27.9 | 15.8 | 12.6 | **63.8** | 21.0 | 6.6 | 75.6 | 12.1 | 5.9 | 77.8 | 11.0 | 2.4 | 70.1 | 4.6 | 12.9 |
| SYNC (Changpinyo et al., 2016) | 7.9 | **43.3** | 13.4 | 11.5 | 70.9 | 19.8 | 8.9 | 87.3 | 16.2 | 10.0 | 90.5 | 18.0 | 7.4 | 66.3 | 13.3 | 16.1 |
| SAE (Kodirov et al., 2017) | 8.8 | 18.0 | 11.8 | 7.8 | 54.0 | 13.6 | 1.8 | 77.1 | 3.5 | 1.1 | 82.2 | 2.2 | 0.0 | 83.3 | 0.0 | 6.2 |
| GFZSL (Verma & Rai, 2017) | 0.0 | 39.6 | 0.0 | 0.0 | 45.7 | 0.0 | 1.8 | 80.3 | 3.5 | 2.5 | 80.1 | 4.8 | 0.0 | 83.3 | 0.0 | 1.7 |
| MIVAE ($\lambda_{map} = 0$) | 20.0 | 40.8 | 26.9 | 27.5 | 55.7 | 36.8 | **36.7** | 84.7 | **51.2** | **38.3** | 87.2 | **53.2** | **16.1** | 85.7 | **27.1** | 39.1 |
| MIVAE ($\lambda_{map} = 1$) | **27.4** | 41.0 | **32.8** | **31.5** | 60.4 | **41.4** | 35.9 | 87.8 | 51.0 | 32.9 | 90.4 | 48.2 | 15.3 | 87.4 | 26.0 | **39.9** |

Figure 3 shows learning curves obtained with these three methods and MIVAE in GZSL. First, for seen classes, all models except MED have nearly equal accuracy. However, because the accuracy of MED is markedly low, it is apparent that manifold learning on latent representation by VAEs is important. In MIVAE, we confirm that the accuracy score is higher when $\lambda_{map}$ is set to 1.

Next, the result of the unseen classes shows that both the MED and the MED-VAE deteriorate considerably during early training. Regarding MED, failure to train generalized representations during training caused bias to seen classes, leading to bad prediction in unseen classes. However, in MED-VAE, accuracy becomes very poor despite manifold training of images, probably because we were unable to train the connection between attributes and images on the embedding space. VZSL tries to link images and corresponding attributes more closely during training, which contributes to making them better than MED-VAE. However, as training progresses, the accuracy drops, i.e., the training becomes unstable.

Finally, it is apparent that MIVAE is markedly more accurate than these methods. From these results, we confirmed that, to predict the unseen classes in the GZSL setting, it might be important to train a shared manifold representation that integrates images and attributes. Moreover, we found that MIVAE can properly obtain such a representation.

The result of the harmonic mean is almost identical to that of the accuracy of unseen classes. In addition, in these AWA2 experiments, the MIVAE's results on accuracy of the unseen classes and the harmonic mean are apparently less dependent on the $\lambda_{map}$ value.

Figure 4 shows representations mapped respectively by the embedding distribution trained with MED-VAE, VZSL, and MIVAE. From this result as well, it is apparent that MIVAE can obtain representations of seen and unseen classes in the latent space.

These results clarify that shared representation learning is insufficient in GZSL and that shared manifold learning is important for high performance.

Table 4: Results in the conventional ZSL setting. We evaluate them by using class-mean top-1 accuracy (%) on unseen classes.

| Methods | SUN | CUB | AWA1 | AWA2 | aPY | Average |
|---|---|---|---|---|---|---|
| CONSE (Szegedy et al., 2015) | 38.8 | 34.3 | 45.6 | 44.5 | 26.9 | 38.0 |
| CMT (Socher et al., 2013) | 39.9 | 34.6 | 39.5 | 37.9 | 28.0 | 36.0 |
| SSE (Zhang & Saligrama, 2015) | 51.5 | 43.9 | 60.1 | 61.0 | 34.0 | 50.1 |
| LATEM (Xian et al., 2016) | 55.3 | 49.3 | 55.1 | 55.8 | 35.2 | 50.1 |
| ALE (Akata et al., 2016) | 58.1 | 54.9 | 59.9 | 62.5 | 39.7 | 55.0 |
| DEVISE (Frome et al., 2013) | 56.5 | 52.0 | 54.2 | 59.7 | **39.8** | 52.4 |
| SJE (Akata et al., 2015) | 53.7 | 53.9 | 65.6 | 61.9 | 32.9 | 53.6 |
| ESZSL (Romera-Paredes & Torr, 2015) | 54.5 | 53.9 | 58.2 | 58.6 | 38.3 | 52.7 |
| SYNC (Changpinyo et al., 2016) | 56.3 | 55.6 | 54.0 | 46.6 | 23.9 | 47.3 |
| SAE (Kodirov et al., 2017) | 40.3 | 33.3 | 53.0 | 54.1 | 8.3 | 37.8 |
| GFZSL (Verma & Rai, 2017) | 60.6 | 49.3 | **68.3** | 63.8 | 38.4 | **56.1** |
| MIVAE ($\lambda_{map} = 0$) | **60.8** | 51.8 | 64.1 | 64.1 | 32.8 | 54.7 |
| MIVAE ($\lambda_{map} = 1$) | 60.2 | **56.5** | 63.1 | **64.7** | 35.4 | 56.0 |

## 5.2 COMPARISON WITH STATE-OF-THE-ART METHODS

Next, we compare MIVAE with state-of-the-art relation-based ZSL. All results of these methods were reprinted from Xian et al. (2018).

As table 3 shows, our proposed method largely outperforms all existing methods of relation-based. Relation-based methods are known to tend to be biased to seen classes, so that the results of unseen classes become considerably worse. Particularly, table 3 shows that performance is poor in datasets with few seen classes such as AWA (both 1 and 2) and aPY. By contrast, despite being a relation-based method, MIVAE can properly generalize unseen classes even in these difficult datasets. These results suggest that our approach solves the bias problem in GZSL. Also, for seen classes, our method is equal to or higher than other methods, so that for harmonic mean, the proposed method is much superior to the others.

In MIVAE, the result of $\lambda_{map} = 1$ is better than that of $\lambda_{map} = 0$ in both SUN and CUB, but there is not much difference between them in terms of AWA and aPY; rather it is better when $\lambda_{map} = 1$.

Note that we do not reprint the results of VZSL in Wang et al. (2017) to table 3 because the train/test split and the evaluation metric in Wang et al. (2017) is different from here. In our reproduction of VZSL in the same split and metric here, the harmonic means in each dataset and their average (SUN, CUB, AWA1, AWA2, aPY, Average) are (20.9, 30.7, 16.7, 18.7, 11.6, 19.7) respectively, which are all inferior to our method.

Finally, we evaluated the performance in the conventional ZSL setting. As table 4 shows, our proposed method has nearly the same performance as that of existing methods in most datasets. These results show that our method is *directly* addressing the bias-related shortcoming of relation-based approaches on GZSL.

## 6 CONCLUSION

In this paper, we addressed the setting of generalized zero-shot learning (GZSL) and specifically demonstrated that shared manifold learning is important to solve the biased problem. Furthermore, to demonstrate that point, we proposed modality invariant variational autoencoders (MIVAE), which realizes shared manifold learning using variational autoencoders that take both images and attributes as inputs. We first compared MIVAE with some ZSL methods that perform only shared representation learning. Thereby, we confirmed that performing manifold learning with both images and attributes is important for good performance of GZSL. Next, we experimented with several benchmark datasets. Results showed that MIVAE has markedly higher performance than state-of-the-art relation-based ZSL methods, which suggested that the bias-related difficulty can be mitigated. Furthermore, the fact that the performance in the conventional ZSL is not much different from that of the existing method means that MIVAE directly confronts difficulties arising with related-based GZSL. Actually, MIVAE is a robust and extensible method for the input information format. Therefore, in the future, we plan to deal with more realistic and complicated images directly and to use text information etc. as more sophisticated side information of classes.

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

## A    PARAMETERIZATION OF DISTRIBUTIONS WITH DEEP NEURAL NETWORKS

The Gaussian distribution can be parameterized with deep neural networks as

$$\mathcal{N}(\boldsymbol{z}; \boldsymbol{\mu}, \mathrm{diag}(\boldsymbol{\sigma}^2)),$$
$$\boldsymbol{\mu} = f_\mu(f_{\mathrm{MLP}}(\boldsymbol{x})),$$
$$\boldsymbol{\sigma}^2 = \mathrm{Softplus}(f_{\sigma^2}(f_{\mathrm{MLP}}(\boldsymbol{x}))),$$

where $f_\mu$ and $f_{\sigma^2}$ are respectively denote linear single layer neural networks and where $f_{\mathrm{MLP}}$ represents a deep neural network with an arbitrary number of layers. Moreover, applying the softplus function for each element of a vector is denoted as $\mathrm{Softplus}$.

The Bernoulli distribution is parameterized as

$$p_\theta(\boldsymbol{x}|\boldsymbol{z}) = \mathcal{B}(\boldsymbol{x}; \boldsymbol{\mu}), \boldsymbol{\mu} = \mathrm{Sigmoid}(f_\mu(f_{\mathrm{MLP}}(\boldsymbol{z}))),$$

where $\mathrm{Sigmoid}$ is represents the sigmoid function.

## B    NETWORK ARCHITECTURES

For the notation of model structures, we denote a linear fully-connected layer with $k$ units, batch normalization, and ReLU as `DkBR`. Also, we denote `DkBR` without batch normalization and ReLU as `Dk`. In addition, the process of applying `J` after `I` is denoted as `I-J`, and the process of concatenating the last layers of the two networks `I, J` into one layer is denoted as `(I,J)`.

Therefore, the network structures of distributions of MIVAE are as follows:

- $p(\boldsymbol{x}|\boldsymbol{z})$ (Gaussian, where $\sigma^2 = 1$)
    - $f_\mu$: `D2048`
    - $f_{\mathrm{MLP}}$: `z-D1000BR`
- $p(\boldsymbol{a}|\boldsymbol{z})$ (Bernoulli)
    - $f_\mu$: `DdimA`
    - $f_{\mathrm{MLP}}$: `z-D1000BR`
- $q(\boldsymbol{z}|\boldsymbol{x}, \boldsymbol{a})$ (Gaussian)
    - $f_\mu$ and $f_{\sigma^2}$: `D1000`
    - $f_{\mathrm{MLP}}$: `(x,a)-D1000BR-D1000BR`
- $q(\boldsymbol{z}|\boldsymbol{x})$ (Gaussian)
    - $f_\mu$ and $f_{\sigma^2}$: `D1000`
    - $f_{\mathrm{MLP}}$: `x-D1000BR-D1000BR`
- $q(\boldsymbol{z}|\boldsymbol{a})$ (Gaussian)
    - $f_\mu$ and $f_{\sigma^2}$: `D1000`
    - $f_{\mathrm{MLP}}$: `a-D1000BR-D1000BR`

## C    DISCUSSION ON PROBABILISTIC MAPPING TO OBTAIN SHARED REPRESENTATION

In shared representation learning on ZSL, we consider nonlinear functions that map each modality as input into the same space $\mathcal{Z}$, i.e. $f_{\boldsymbol{x}} : \mathcal{X} \to \mathcal{Z}$ and $f_{\boldsymbol{a}} : \mathcal{A} \to \mathcal{Z}$.

If each mapping can be trained so that the same representation can be extracted from each modality of the same example, i.e. $\forall i \boldsymbol{z}_{xi} = \boldsymbol{z}_{ay_i}$, where $\boldsymbol{z}_{xi} = f_{\boldsymbol{x}}(\boldsymbol{x}_i)$ and $\boldsymbol{z}_{ay_i} = f_{\boldsymbol{a}}(\boldsymbol{a}_{y_i})$, we should be able to predict labels of the images in the test set by selecting the class that corresponds to the closest representation, i.e. $\hat{y}_j = \arg\min_{y \in \mathcal{Y}} dis(\boldsymbol{z}_{xj}, \boldsymbol{z}_{ay})$, where $dis(a, b)$ is an arbitrary distance function between $a$ and $b$. Therefore, the simplest approach is to train the mappings so that the distance of the representations of the two modalities is small on the training set, i.e. $\min_{f_{\boldsymbol{x}}, f_{\boldsymbol{a}}} \sum_{i=1}^{N_{tr}} dis(f_{\boldsymbol{x}}(\boldsymbol{x}_i), f_{\boldsymbol{a}}(\boldsymbol{a}_{y_i}))$.

However, this method is highly likely to fail for the following two reasons. First, although only one example of attributes corresponding to a class exists, there are countless examples of images corresponding to that class. Therefore, if we attempt to make representations of attributes and images contained in one class closer, then these representations will degenerate to one point. Another reason is that if one can train to correspond well between each modality of training data, then that representation might not generalize to test data.

To solve these problems, we consider the uncertainty of each example of each modality. To account for this uncertainty, we replace the deterministic embedding functions, $f_{\boldsymbol{x}}$ and $f_{\boldsymbol{a}}$ with the deep probabilistic distributions, $p(\boldsymbol{z}_i|\boldsymbol{x}_i)$ and $p(\boldsymbol{z}_i|\boldsymbol{a}_{y_i})$.

The benefits of considering distributions for each example are twofold. First, we can prevent the degeneracy of representation which occurs when making examples of modalities that do not closely correspond one-to-one. This is true because we can consider the distance between distributions of each example, not between each example. Second, minimizing the distance between distributions of examples of each modality implies that the distribution of each modality is closer, i.e., $\forall i \in \{1..., N_{tr}\}[p(\boldsymbol{z}_i|\boldsymbol{x}_i) = p(\boldsymbol{z}_i|\boldsymbol{a}_{y_i})] \Rightarrow p(Z_x) = p(Z_a)$, where $Z_x = \{\boldsymbol{z}_{x1}..., \boldsymbol{z}_{xN_{tr}}\}$ and $Z_a = \{\boldsymbol{z}_{ay_1}..., \boldsymbol{z}_{ay_{N_{tr}}}\}$. This point might be readily apparent from $\prod_{i=1}^{N_{tr}} p(\boldsymbol{z}_i|\boldsymbol{x}_i) = p(Z|X) = p(Z_x)$. Therefore, if we can train to bring each distribution closer, then we can obtain shared representation of them.

