# OpenReview forum: "Learning shared manifold representation of images and attributes for generalized zero-shot learning"
_ICLR.cc/2019/Conference_

### Official Review · AnonReviewer1 · 2018-10-31
**Good Generalized Zero-shot learning experimental results but limited contributions**

**Rating:** 5
**Confidence:** 4

**Review:**

The paper considers the problem of (Generalized) Zero-Shot Learning. Most zero-shot learning methods embed images and text/attribute representations into a common space. The main difference here seems to be that Variational AutoEncoder (VAEs) are used to learn the mappings that take different sources as input (images and attributes).
As in JMVAE (Suzuki et al., 2016) (which was not proposed for zero-shot learning), decoders are then used to reconstruct objects from the latent space to the input sources.

My main concerns are about novelty. The contribution of the paper is limited or not clear at all, even when reading Section C in the appendix. The proposed approach is a straightforward extension of JMVAE (Suzuki et al., 2016) where a loss function is added (Eq. (3)) to minimize the KL divergence between the outputs of the encoders (which corresponds to optimizing the same problem as most zero-shot learning approaches).
The theoretical aspect of the method is then limited since the proposed loss function actually corresponds to optimize the same problem as most zero-shot learning approaches but with VAEs.

Concerning experiments, Generalized Zero shot learning (GZSL) experiments seem to significantly outperform other methods, whereas results on the standard zero-shot learning task perform as well as state-of-the-art methods.
Do the authors have an explanation of why the approach performs significantly better only on the GZSL task?

In conclusion, the contributions of the paper are mostly experimental. Most arguments in the model section are actually simply intuitions.


after the rebuttal:
After reading the different reviews, the replies of the authors and the updated version, my opinion that the "explanations" are simply intuitions (which is related to AnonReviewer3's concern "Regarding advantages of learning a joint model as opposed to unidirectional mappings") has not been completely addressed by the authors. Fig. 4 does address this concern by illustrating their point experimentally. However, I agree with AnonReviewer3 that the justification remains unclear.

---

> ### Author Response · Authors · 2018-11-26
> **Reply to Reviewer 1**
>
> Thank you very much for informative comments and sorry for my late reply.
>
> As you pointed out, in terms of learning the shared representation of different modalities, the proposed method is considered to be almost the same as existing relation-based GZSL. However, the most important difference is that our method performs manifold learning on the shared representation using VAEs.
>
> In GZSL, the test data also includes the seen classes, so it is necessary for both examples of the seen classes and the unseen classes to be properly placed on the shared latent space. However, in the conventional relation-based method, they could not successfully map the examples of the unseen classes in test data to the shared representation because this mapping tends to degenerate or overlap with the seen classes. In other words, this representation did not "generalize" to the unseen classes. Therefore, even though the accuracy of the seen classes is high, the accuracy of the unseen classes results in very low.
>
> On the other hand, MIVAE proposed in this paper performs manifold learning on shared representations that integrate the two modalities, which means that the position of the unseen classes in the latent space can be estimated by "interpolating" from the training data (seen classes). Therefore, the problem of degeneration and overlapping with seen class is resolved, which improves the accuracy of the unseen classes.
>
> Moreover, please note that in conventional ZSL, since the seen classes are not included in the test dataset, there is no problem that the mapping to the latent space overlaps with the seen classes at the testing time. Therefore, the performance of MIVAE does not differ much from the existing method in conventional ZSL.
>
> In summary, this research resolves the problem that the mapping of the unseen classes in GZSL is not well arranged, by manifold learning on shared representation. As you pointed out, MIVAE itself is a straightforward extension of JMVAE, but we believe that our research has novelty in the sense that it first showed that the method using VAEs is effective in GZSL.

---

### Official Review · AnonReviewer3 · 2018-11-02
**Interesting and novel extension of joint-VAEs in zero-shot learning. Could be written with more clarity and justification regarding some design choices.**

**Rating:** 5
**Confidence:** 4

**Review:**

The paper proposes an approach to generalized zero-shot learning by learning a shared latent space between the images and associated class-level attributes. To learn such a shared latent space and mapping for the same which is generalized and robust -- the authors propose ‘modality invariant variational autoencoders’ -- which allows one to perform shared manifold learning by training VAEs with both images and attributes as inputs. Empirical results demonstrate improvements over existing approaches on the harmonic mean metric present in the generalized zero-shot learning benchmark. Other than the concerns mentioned below, I like the basic idea adopted in the paper to extend Vedantam et. al. (2018)’s joint-VAEs (supporting unimodal inference) to the framework of generalized zero-shot learning. The proposed approach clearly results in improvements over baselines and existing approaches.

Comments:
- A minor correction. The paper claims the bias towards seen classes at inference for the existing GZSL approaches is due to the inability of obtaining training data for the unseen classes. In my opinion, this should be rephrased as the inability to learn a generalized enough representation (joint or otherwise) that is aware of the shift in distribution from seen to unseen classes (images or attributes) as this information is not available apriori.
- Writing Clarity Issues. In general, there is significant repetitions along certain lines throughout the introduction and approach. While the paper overall does a good job of explaining the motivation as well as the approach, some of the sections (and sentences within) could be written better to express the point being made. Specifically, the first paragraph in the introduction seems to be structured more from a few-shot setting. The paper would benefit from talking about few-shot learning first and then extending to the extreme setting of zero-shot learning. Similarly, the second paragraph in the introduction could be written more succinctly to express the point being made. The sentences -- “Moreover, it is difficult….widely available” -- are difficult to understand. Tables 4 and 5 should be positioned after the references section.
- A point repeatedly made in the paper suggests that learning unidirectional mappings from images to attributes (or otherwise) suffers from generalization to unseen classes. While I agree with this statement, most methods in GZSL hold out a subset of seen classes as validation (unseen) classes while learning such a mapping -- which I believe was also being done while learning the joint model in MIVAE (Can the authors confirm this? Is yes, how were these classes chosen?). As such, the authors should stress on the advantages learning a joint latent model over both modalities offers as opposed to unidirectional mappings while mentioning the above points.
- Learning the joint latent space for images and attributes has been referred to as learning a shared manifold in the paper -- with associated terms such as manifold representation being used as well. Sharing a latent space need not imply learning an entire manifold as the subspace captured by the latent space might as well be localized in the manifold in which it exists. Can the authors comment more on this connection with respect to the points around “shared manifold learning”?
- During inference, the authors operate in the latent space to find the most-relevant class by enumerating over all classes the KL-divergence between the unimodal encoder embeddings. Is there a particular reason the authors chose to operate in the latent space as opposed to operating in a modality space? Specifically, given an image the authors could have used the p(a|z) decoder to infer the attribute given the encoded z -- and subsequently finding the 1-nearest neighbor in that space. Any reason why this approach was not adopted?
- On page 4, regarding the term L_dist in the objective for MIVAE, the authors draw the connections made in the appendix of Vedantam et. al. (2018) regarding the minimization of KL-divergence between the bimodal and a unimodal variational posterior(s). While the connection being made is accurate, the subsequent solution modes identified in the following paragraph -- “When equation 2 becomes minimum…” -- do not seem accurate. At minimality, unimodal encoders should be equivalent to the bimodal encoder marginalized over the absent rv under the conditional distribution of the data. Could the authors comment on whether the version presented in the paper is intended or is merely a typographical mistake?
Section 5 experiments suggest the learning rate used in practice was 10^3. Assuming a typo, this should be presumably 10^-3.

Experimental Issues.
- The authors should explicitly mention if they are using the proposed split throughout all baselines and approaches for GZSL evaluations. It’s not explicitly mentioned in the text and is an important detail that should not be left out. Only the appendix mentions the number of seen/unseen classes.
- How did the authors select a validation split (held out seen classes) to train MIVAE? Did they directly borrow the training and validation splits present in the proposed split? Or did they create a split of their own? If latter, how was the split created? In general, I am curious about how the MIVAE checkpoint for inference was chosen.
- In section 5.2, the reasons in the 3rd paragraph elaborating \lambda_map=1 vs 0 not being too different for AWA and aPY are not clear. Could the authors comment a bit more on them?

The authors adressed the issues raised/comments made in the review. In light of my comments below to the author responses -- I am not inclined towards increasing my rating and will stick to my original rating for the paper.

---

> ### Author Response · Authors · 2018-11-27
> **Reply to Reviewer 3 (1/2)**
>
> Thank you very much for positive comments and apologize for our late reply.
>
> >> While the paper overall does a good job of explaining the motivation as well as the approach, some of the sections (and sentences within) could be written better to express the point being made.
> Thank you for pointing it out. We have modified several sentences to make it as readable as possible throughout this paper.
>
> >>  A minor correction. The paper claims...
> >>  Specifically, the first paragraph in the introduction seems to be structured more from a few-shot setting.
> >>  Similarly, the second paragraph in the introduction could be written more succinctly to express the point being made.
> >> The sentences -- “Moreover, it is difficult….widely available” -- are difficult to understand.
> >> Tables 4 and 5 should be positioned after the references section.
> Thank you for pointing out in detail. These have been fixed in the current version, so we would be grateful if you could check them.
>
> >> As such, the authors should stress on the advantages learning a joint latent model over both modalities offers as opposed to unidirectional mappings while mentioning the above points.
> Following your comment, we modified section 3.1 and added the advantages of learning on shared representation. Below we will briefly explain these advantages.
>
> Firstly, shared representation is rich compared to attribute representation because it integrates both image and attribute. In attribute space, the performance of GZSL is significantly reduced unless each class is properly represented as attributes in advance (Chao et al. 2016). On the other hand, shared representation is not only richer than a representation of attributes but also robust against it. Therefore, in the shared space, relations between modalities are learned more properly than attribute space.
>
> Another advantage is that since shared space does not depend on input dimensions or representation, we can perform zero-shot learning with more complex and sophisticated inputs.
>
> In this study, we did not monitor learning using validation data but used all of them as training data during training. To our knowledge, this is also done in other GZSL studies (Verma, V. Kumar, et al., 2018).  This is because in GZSL the test data also includes the seen classes, so if we do not use a part of the seen classes for training, the performance of seen data in the test set might be decreased. In all experiments, the training is terminated with 200 epochs.
>
> >> - Learning the joint latent space for images and attributes has been referred to as learning a shared manifold in the paper...
> In this paper, we referred to learning the representation that generalizes to the unseen classes as "manifold learning". The reason for this is because we wanted to differentiate from learning relationships between both modalities. Moreover, "shared manifold learning" refers to learning to generalize both relationships between modalities and separation between classes.
>
> Actually, if manifold learning is properly performed in the shared space, the position of the arbitrary unseen class in the shared space should be obtained by interpolating the representation of the seen classes. However, it may not be very accurate to use the word "manifold learning" in order to refer to such things.
>
> >> - During inference, the authors operate in the latent space...
> As explained above, since shared space is more rich representation than attribute space, we thought that we can obtain generalized relations in shared space with higher accuracy. In a simple experiment in ZSL, we confirmed that the case of shared space has a higher performance than that of the attribute space.
>
> >> - On page 4, regarding the term L_dist in the objective for MIVAE...
> As you pointed out, this was our mistake. In the current version, we modified this sentence.
>
> >> Section 5 experiments suggest the learning rate used in practice was 10^3. Assuming a typo, this should be presumably 10^-3.
> Thank you for pointing it out. We fixed 10^3 to 10^-3.

---

> > ### Author Response · Authors · 2018-11-27
> > **Reply to Reviewer 3 (2/2)**
> >
> > >> - The authors should explicitly mention if they are using the proposed split throughout all baselines and approaches for GZSL evaluations.
> > Following your comment, we added a description about the split of datasets.
> >
> > >> - In section 5.2, the reasons in the 3rd paragraph...
> > We think that it relates to the number of training data of each dataset.  In order to learn the generalized relations of different modalities, a sufficient amount of data is needed. However, in SUN and CUB, there are not much training data compared with the number of their classes, so it is difficult to learn relations between modalities. Therefore, it is considered that the terms explicitly bringing the relation closer (equation 3) contributed to the improvement of their performance. On the other hand, since AWA and aPY have relatively sufficient data, it is considered that equation 2 could be properly learned without adding equation 3.
> >
> > In any case, we believe that we need to further verify this phenomenon in more detail.

---

> > > ### Comment · AnonReviewer3 · 2018-11-29
> > > **Comments regarding the Author Response**
> > >
> > > Thanks to the authors for providing detailed comments and explanations wherever applicable and revising the paper to reflect the same as well. In light of the comments below (regarding author responses to review comments and clarifications), I am slightly skeptical of the results presented in the paper. As such, I will be sticking to my original rating for the paper.
> > >
> > > - Edits with respect to the GZSL-bias discussion: Thanks for clarifying the seen-unseen bias issue in detail. I will merge 2 discussion points here and highlight what is missing. Firstly, the authors clarified that they did not hold out a set of seen classes as validation to test generalization during training the joint model. Assuming there is no access to the test-split during training, absence of a validation set implies the choice of hyper-parameters for training MIVAE was not done in a manner to test generalization. As such, an arbitrary choice of stopping training at 200 epochs across all experiments (assuming this is across datasets) is odd. The validation subset of seen classes in itself could be used to perform hyper-parameter sweeps -- the resulting values from which could be used to train the entire setup again on the entirety of the seen training set. Secondly, given this fact, it seems that no notion of generalization to an unseen split (even for the choice of hyper-params) was done and hence from Section 3.1 (and other sections) it is unclear to me how the “shared representation” from the proposed approach could help in alleviating the bias issue. In general, I am now curious about how the hyper-parameters were chosen -- I think this clarification is quite important. The paper the authors point to w.r.t. this issue does indeed use a validation split if I’m correct -- see Section 6.1, line - “The hyper-parameters were chosen … were used while training the model on complete data.” of https://arxiv.org/pdf/1712.03878.pdf.
> > >
> > > - Regarding advantages of learning a joint model as opposed to unidirectional mappings: Thanks for responding to this comment in detail. However, the response still does not explicitly state why learning a joint model -- that allows one to do inference from attributes to images, images to attributes and latent variable to either modalities -- is better than just learning a single unidirectional mapping from attributes to images or images to attributes. I agree that the shared representation is richer -- but it is not convincing enough as to why is this needed in the context of GZSL in text? The authors should talk more about the experiment they refer to regarding inference in latent space regarding this.

---

> > > > ### Author Response · Authors · 2018-12-07
> > > > **Thank you for your comment**
> > > >
> > > > Thank you very much for your comment on our reply. We would like to respond to your questions as much as possible.
> > > >
> > > > >> Edits with respect to the GZSL-bias discussion
> > > > First of all, we apologize for having given you an answer that confused you. To determine the hyper-parameters (including the number of epochs), as you mentioned, we did use a part of the classes in the training set as validation data for "zero-shot" validation  (this is written in our updated paper in detail, so please see section 5). Therefore, we would like to emphasize that hyper-parameters were not arbitrarily selected from the results on the test set. Furthermore, in our experiments, we confirmed that even if the number of epochs is large, the classification accuracy of our model does not decrease so badly (but I'm sorry that this paper includes only results up to 50 epochs).
> > > >
> > > > On the other hand, *after* setting hyper-parameters, we used *all* of the training set for the training. In other words, in order to *monitor* whether the model is generalized during training, we did not prepare the validation data, which is what we wanted to answer in our previous reply and which is also done in other GZSL studies. Also please note that we cannot check whether the GZSL-specific biased problem is occurring during hyper-parameter setting and training due to "zero-shot" validation. It may be possible to monitor the biased problem during training by "generalized zero-shot" validation, i.e., by using part of all seen classes (other than validation classes) as validation data, too. However, this has a problem that the number of data for training is further reduced. In our study, we proposed to perform manifold learning as a way to prevent unseen classes from overlapping the seen classes in latent space. As you pointed out, there is no theoretical guarantee that this method always generalizes to unseen classes, but considering the above problems and our results, we believe that our method has effectiveness.
> > > >
> > > > >> Regarding advantages of learning a joint model as opposed to unidirectional mappings
> > > > The problem of attribute space is that the placement within the space of the classes is pre-defined so that it may not be well separated between classes when the attribute representation is bad, which is also pointed out in Chao et al. (2016). On the other hand, since there is no bias due to such pre-defined in image representation, if manifold learning of images is properly performed, it can obtain a well-separated representation between classes. Furthermore, if attributes are also used as inputs, more separated representation might be obtained. We thought that if we could acquire such representation (shared manifold representation), we could map the unseen data to places that do not overlap with the training set (seen data) and solve the biased problem. However, this insight is empirical and, as you pointed out, we should have included the results of comparison with attribute space in the paper.

---

### Official Review · AnonReviewer2 · 2018-11-02
**application of multimodal VAE for zero shot learning**

**Rating:** 4
**Confidence:** 5

**Review:**

This paper proposes a multimodal VAE model for the problem of generalized zero shot learning (GZSL). In GZSL, the test classes can contain examples from both seen as well as unseen classes, and due to the bias of the model towards the seen classes, the standard GZSL approaches tend to predict the majority of the inputs to belong to seen classes. The paper proposes a multimodal VAE model to mitigate this issue where a shared manifold learning learn for the inputs and the class attribute vectors.

The problem of GZSL is indeed important. However, the idea of using multimodal VAE for ZSL isn't new or surprising and has been used in earlier papers too. In fact, multimodal VAEs are natural to apply for such problems. The proposed multimodal VAE model is very similar to the existing ones, such as Vedantam et al (2017), who proposed a broad framework with various types of regularizers in the multimodal VAE framework. Therefore, the methodological novelty of the work is somewhat limited.

The other key issue is that the experimental results are quite underwhelming. The paper doesn't compare with several recent ZSL and GZSL approaches, some of which have reported accuracies that look much better than the accuracies achieved by the proposed method. The paper does cite some of these papers (such as those based on synthesized examples) but doesn't provide any comparison. Given that the technical novelty is somewhat limited, the paper falls short significantly on the experimental analysis.

---

> ### Author Response · Authors · 2018-11-26
> **Reply to Reviewer 2**
>
> Thank you very much for your valuable feedback and sorry for our late response.
>
> >> However, the idea of using multimodal VAE for ZSL isn't new or surprising and has been used in earlier papers too.
> VZSL (Wang et al., 2017) cited in our paper is known as an example of a study applying VAE to ZSL, but to the best of our knowledge, there are not many studies applying multimodal VAE to ZSL.
>
> >> The proposed multimodal VAE model is very similar to the existing ones, such as Vedantam et al (2017), who proposed a broad framework with various types of regularizers in the multimodal VAE framework.
> As you pointed out, Vedantam et al. (2017) uses multimodal learning using two modalities, attributes and images, and generates unseen images from corresponding attributes. However, this work is not intended to solve the problem of zero-shot learning, because it does not predict the class labels of unseen classes.
>
> In addition, rather than introducing a completely novel model, we showed that manifold learning in shared space using VAEs is effective to resolve a problem of relation-based GZSL. As written in our paper, the conventional relation-based GZSL had an inherent problem of failing to predict the unseen classes in the test data (please note that this problem only occurs in GZSL, where the test class contains the seen class). This is because the mapping to the shared space of the unseen classes does not generalize well and overlaps with the seen classes, which we call the biased problem. In this study, we showed that manifold learning using VAEs can appropriately place unseen classes in shared space by interpolating from seen classes.
>
> From the results in table 3, the accuracy of the proposed method was improved significantly in the unseen classes while the accuracy does not change very much in the seen classes, which means that that the proposed method resolves the biased problem directly. To our knowledge, there is no other study showing that this problem in relation-based is improved.
>
> >> The paper doesn't compare with several recent ZSL and GZSL approaches, some of which have reported accuracies that look much better than the accuracies achieved by the proposed method.
> As mentioned above, our study focused on the bias problem of relation-based GZSL and proposed MIVAE as a method to solve it, which is the main contribution of this paper. Therefore, in our experiments, we compared the proposed method with relation-based studies which have the biased problem in order to confirm whether the problem was resolved. This is the reason why we did not compare it to synthesis-based methods in this paper.
>
> Furthermore, as we wrote in the section of related works, synthesis-based needs to generate images from attributes, so it might be difficult to improve accuracy if attributes or images become complicated. On the other hand, the relation-based method (shared representation method in particular) allows us to select class-attributes closest to the given image in the space of shared representation, which means that we can perform ZSL without depending on the complexity of the input information.
> Our method solved the inherent bias problem while maintaining this advantage of relation-based, so we believe that our method is highly extensible.

---

### Meta-Review · Area_Chair1 · 2018-12-13

**Confidence:** 5
**Recommendation:** Reject

**Metareview:**

The paper addresses generalized zero shot learning (test data contains examples from both seen as well as unseen classes) and proposes to learn a shared representation of images and attributes via multimodal variational autoencoders.
The reviewers and AC note the following potential weaknesses: (1) low technical contribution, i.e. the proposed multimodal VAE model is very similar to Vedantam et al (2017) as noted by R2, and to JMVAE model by Suzuki et al, 2016, as noted by R1. The authors clarified in their response that indeed VAE in Vedantam et al (2017) is similar, but it has been used for image synthesis and not classification/GZSL. (2) Empirical evaluations and setup are not convincing (R2) and not clear -- R3 has provided a very detailed review and a follow up discussion raising several important concerns such as (i) absence of a validation set to test generalization, (ii) the hyperparameters set up; (iii) not clear advantages of learning a joint model as opposed to unidirectional mappings (R1 also supports this claim). The authors partially addressed some of these concerns in their response, however more in-depth analysis and major revision is required to assess the benefits and feasibility of the proposed approach.